# Seasonal variations in the nutritive value of fifteen multipurpose fodder tree species: A case study of north-western Himalayan mid-hills

Manasi Rajendra Navale[1], D. R. Bhardwaj[1]*, Rohit Bishist[1], C. L. Thakur[1], Subhash Sharma[2], Prashant Sharma[1]*, Dhirender Kumar[1], Massimiliano Probo[3]

1 Department of Silviculture and Agroforestry, Dr. Y.S.P. University of Horticulture and Forestry, Solan, India, 2 Department of Social Sciences, Dr. Y.S.P. University of Horticulture and Forestry, Solan, India, 3 Grazing Systems, Agroscope, Posieux, Switzerland

* bhardwajdr_uhf@rediffmail.com (DRB); prashantsharma92749@gmail.com (PS)

**Data Availability Statement:** All relevant data are within the paper and its Supporting Information files.

## Abstract

Multipurpose tree species are recognized as an important fodder source for livestock, but their potential remains untapped due to dearth of knowledge about their nutritive value. Therefore, 15 MPTs, i.e., *Acacia catechu*, *Albizia chinensis*, *Bauhinia variegata*, *Celtis australis*, *Ficus roxburghii*, *Grewia optiva*, *Leucaena leucocephala*, *Melia composita*, *Morus serrata*, *Olea glandulifera*, *Ougienia oojeinensis*, *Pittosporum floribundum*, *Quercus glauca*, *Q. leucotrichophora* and *Salix tetrasperma* were evaluated for nutritional characteristics, relative nutritive value index (RNVI), palatability index and farmers' preference on a seasonal basis in north-western Himalayas mid-hills. Most of the nutritive and mineral content decreased as leaves matured with the exception of ether extract, calcium, copper, organic matter and carbohydrate content, while cell-wall constituents and anti-nutritional contents increased. Overall, *M. serrata* had the highest RNVI in spring and summer, while *G. optiva* during autumn and winter. Similarly, *L. leucocephala* had the highest palatability (97.86%), while *M. composita* (38.47%) had the lowest one. Additionally, *G. optiva* was the most favored MPT for livestock among farmers, while *M. composita* was the least ones. The outcome of the study will help policy makers, planners and farm managers in establishing large scale plantations of highly nutritious and palatable species, like *G. optiva*, *L. leucocephala*, *B. variegata*, and *M. serrata* for year-round supply of green leaves and as a supplement to low-quality feed.

## Introduction

Livestock are global assets and one of major component of agricultural sector enabling many smallholders in India to escape poverty. India has the world's largest livestock population, with 538.8 million livestock in 2019, increased by 4.6 per cent in comparison to 2012 [1] and contributing to 5.1 and 17.1 per cent of the total gross value added (GVA) and agriculture & allied

**Funding:** The authors received no specific funding for this work.

**Competing interests:** The authors have declared that no competing interests exist.

sector GVA, respectively [2]. Obviously, this trend of rising livestock population has an effect on the kind of fodder resources required to satisfy the nutritional requirements [3]. Indeed, the nutritional health of livestock is crucial for a sustainable production, which depends on food availability, animal nutrient needs, feed nutritional quality, consumption, digestibility [4] as well as feed metabolism [5]. However, in India, either the natural pastures are degraded or have the carrying capacity of less than one livestock unit per hectare per year [3]. Simultaneously, it becomes difficult to grow forage crops on farms due to exponential increase of human population and demand for food grains [6].

Therefore, the scarcity of conventional feeds for cattle has forced nutritionists to explore for alternative feed sources [7], when forage sources such as legumes are expensive [8]. This is the case of leaf fodder from multipurpose trees species (MPTs), which are widely regarded as emergency feedstuffs [9], especially in the hilly region [10], due to the high protein, soluble carbohydrate, mineral, and vitamin content of their leaves [11, 12]. For this reason, MPTs are left deliberately or planted on farm bunds [13], since they can provide versatile products and play a significant role in rural economies [14, 15]. Therefore, MPTs have a fair chance of increasing livestock productivity by reducing the gap between demand and availability of green fodder [16]. In addition, MPTs develop deep tap root system, which can maintain the green phytomass late in the season when the herbaceous layer is dry [16, 17].

Concurrently, several variables, including inherent species genotype [18], seasonal variance [19–21], site capacity, edapho-climatic factors [22] and management aspects [23], contribute in determining the nutritional quality dynamics of fodder trees. Among these, temporal fluctuation in the chemical composition and digestibility of tree species may occur as a result of phenological and climatic variations [21]. Thus, knowledge of tree foliage mineral and nutrient fluctuations during the seasons offers considerable potential to provide high-quality ruminant forage [19], by selecting an appropriate harvesting period for specific tree species in a particular agro-climatic region [24]. Such awareness contributes to their proper usage, as well as to the detection of nutrient deficiencies and recommendations for additional livestock requirements [25].

Moreover, indigenous forest trees and shrubs have recently received considerable attention in research [26]. Indeed, local farmers generally have invaluable traditional knowledge of indigenous and exotic MPTs, but little idea about their nutritional values. Therefore, farmer engagement is crucial, as their knowledge and preferences as future users are essential [27]. Additionally, the integration of scientific and farmer's knowledge is critical, because it is often not practical to collect the information on the chemical composition of different MPTs on a consistent basis, due to the large number of fodder trees species used by farmers. Simultaneously, the complementarities between the two information systems will serve as a catalyst for a more integrated approach to evaluate and select the most suitable tree fodder species for their needs [28].

The nutritional value of some commercial MPTs forage has been studied extensively [13, 16, 29], however scarce data are available on less commercially important but common MPTs, in particular with regard to changes in forage quality due to seasonal variation and subsequent palatability in the north-western Himalayas. Therefore, to fill this knowledge gap, we selected fifteen different MPTs which were harvested during the four seasons. The objective of the work was to assess and compare the proximate and mineral composition, anti-nutritional and cell wall components, relative nutritive value and farmers' preference of the selected trees species.

## Materials and methods

### Study area

The study area located in the mid-hill zones of the north-western Himalayas in India (30˚ 51´ N, 76˚ 11´ E, elevation 1250 m above mean sea level), having undulating and hilly terrain with

elevation and depressions, gentle slopes, and south-eastern aspect. The area comes under the sub-tropical belt, but slightly skewed towards the temperate climate, with temperatures ranging from 1˚C in winter to 37˚C during the summer, with a mean annual temperature of 19.8˚C. The hottest months are May and June, while the coldest months are December and January. The area receives 1100–1150 mm of rain per year, with most of the rain falling during the monsoon season (July and August) and rarely experiencing snowfall [30, 31]. The soil is gravelly sandy loam (Order Inceptisol; Typic Eutrochrept) with 62.9% sand, 22.3% silt, and 14.8% clay with neutral soil reaction, low in organic carbon content (0.13–0.19%), medium in available N (300–360 kg ha$^{-1}$) and available K content (300–370 kg ha$^{-1}$), while contains high available P content (40–50 kg ha$^{-1}$) [32].

## Leaf sampling and pre-processing

Leaves of fifteen MPTs were randomly collected from farm fields (naturally grown on the farm bunds), from March 2014 to February 2015 based on the leaf phenology (Table 1) from all parts of the tree crown (three trees per season per species per replicate). During July and August months (rainy season) green fodder is available in plenty and easily, therefore tree species are not fed to animals and hence omitted for current evaluation. For each individual MPTs, collected samples were washed, dried (60±5˚C heating air burning until constant weight was obtained), grounded (the Willey mill) and sieved through a 40- mesh sieve in the laboratory for proximate and mineral content analysis.

## Chemical analysis

The pre-processed leaves samples were analyzed for the proximate principles [33], i.e., dry matter (DM) (%), crude protein (CP) (%), crude fiber (CF) (%), ether extract (EE) (%), total ash (%), nitrogen free extract (NFE) (%), total carbohydrate (%), organic matter (%), and the cell wall constituent [acid detergent fiber (ADF) and neutral detergent fiber (NDF) (%)] [34–36] (S1 File). The flame photometer method [37, 38] was used to determine the phosphorus (P), potassium (K), and calcium (Ca) content while Atomic-absorption spectrophotometer method [39] for the copper (Cu), iron (Fe), zinc (Zn) and manganese (Mn) content. For mineral analysis, the samples were digested using the diacid (HNO3 + HClO4) in the ratio of 4:1. The total phenol (Folin-Ciocalteau reagent method [40]), tannin [41, 42], nitrate [43], hydrocyanic acid (HCN) (talkaline-titration method, [33]) and saponin contents [44] were also assessed. The mimosine content, an alleo-chemical found in the leaves of the *L. leucocephala*, was assessed according to the procedure of Matsumato and Sharman [45].

## Relative Nutritional Value Index (RNVI)

A seasonal relative nutritional value index (RNVI) was created to rank the various MPTs in terms of their nutritive value. The MPTs with the highest value of desirable nutritional traits, i.e., CP, EE, total ash, NFE, OM, total carbohydrate, P, K, Ca, Cu, Fe, Mn and Zn, were assigned a score of (+) 10 for a given trait, while the highest values for traits such as CF, ADF, NDF and anti-nutritional traits, i.e., total phenol, tannin, nitrate, HCN and saponin content were given a score of (-) 10. The remaining species were weighted based on the ratio between their contents and the highest values found for a given trait. The scores obtained by each MPTs were then summed to rank the MPTs.

## Palatability analysis

The cafeteria technique suggested by Larbi et al. [46] and later adopted by Mokoboki et al. [47] was used for the palatability analysis at the Dairy farm of the Department of Silviculture and

**Table 1. Description of different MPTs of mid-hills of north-western Himalayan ecosystem with their leaf phenology.**

| Species | Family | Common name | Nature | Uses | Average leaf dry biomass yield (kg DM tree$^{-1}$ yr$^{-1}$) | Leaf phenology | | | |
|---|---|---|---|---|---|---|---|---|---|
| | | | | | | Spring | Summer | Autumn | Winter |
| *Acacia catechu* Willd. | Fabaceae | Khair | Deciduous | Fodder, fuel, dye, timber, tannin, gum resin | 3.34 | oldest | leafless | new | old |
| *Albizia chinensis* Osbeck. Merr. | Fabaceae | Chinese albizzia | Deciduous / Evergreen | Fodder, fuel, timber, gum resin, erosion control, reclamation | 5.85 | new | old | older | oldest |
| *Bauhinia variegata* Linn. Vern | Fabaceae | Kachnar | Deciduous | Fodder, food, fibre, apiculture fuel, dye, timber, tannin, gum resin, medicinal, ornamental | 7.92 | leafless | new | old | leafless |
| *Celtis australis* Linn. | Ulmaceae | Khirak | Deciduous | Fodder, fuel, fibre, timber, Nitrogen fixing, | 5.18 | new | old | oldest | leafless |
| *Ficus roxburghii* Wall. | Moraceae | Timbal | Evergreen | Food, fodder, stem's latex for cuts & wounds | 5.18 | new | old | older | oldest |
| *Grewia optiva* J. R. Drummond ex Burret. | Tiliaceae | Bhimal | Deciduous | Fodder, fuel, fibre, timber | 2.77 | oldest | leafless | new | old |
| *Leucaena leucocephala* (Lam.) De Wit. | Fabaceae | Subabul | Evergreen | Fodder, fuel, fibre, timber, tannin, dye | 4.28 | oldest | new | old | older |
| *Melia composita* Wild. | Meliaceae | Darek | Deciduous | Fodder, fuel, timber, medicine, Ornamental, beads and rosaries made from fruit beads | 13.30 | leafless | new | old | leafless |
| *Morus serrata* Roxb. | Moraceae | Himalayan mulberry | Deciduous | Food. fibre, fuel, fodder, tannin dye, essential oil, medicine | 6.75 | new | old | oldest | leafless |
| *Olea glandulifera* Wall. Ex G. Don | Oleaceae | Jharinu | Evergreen | Fodder, species coppices well | 8.52 | new | old | older | oldest |
| *Ougienia oojeinensis* (Roxb.) Hochr | Papilionoideae | Sandan | Deciduous | Fodder, fuel, fibre, timber, host plant for lac insects | 12.15 | leafless | new | old | oldest |
| *Pittosporum floribundum* Wight & Arn | Pittosporaceae | Pipalu | Evergreen | Fodder, bark is medicinal contains saponins and pittosporin | 15.65 | older | oldest | new | old |
| *Quercus glauca* Thunb | Fagaceae | Bani oak | Evergreen | Fodder, Timber, tannin, medicine, ornamental | 51.51 | oldest | new | old | older |
| *Q. leucotrichophora* A. Camus | Fagaceae | Ban | Evergreen | Fodder, Timber, medicine, ornamental | 9.56 | older | oldest | new | old |
| *Salix tetrasperma* Roxb. | Salicaceae | Indian willow | Deciduous | Fodder, basket work twigs, construction & planking wood. | 6.59 | new | old | oldest | leafless |

Spring: March-April; Summer: May-June; Autumn: September-November; Winter: December-February

Agroforestry, Dr Y.S.P. University of Horticulture and Forestry, Nauni, India. Six heifers (1–2 years) of Jersey cross breed were chosen and adapted to the selected species by feeding them for about 5 days before starting actual investigation. Two kg of fodder per animal per day from a particular MPT were fed to heifers and on every alternate day, and the same heifer was fed with fodder from another MPT every other day to avoid habituation. This approach was repeated for the six days and for all selected MPTs during their respective fodder production month (S1 and S2 Tables). Furthermore, the animals were fed at 10 a.m., and the final readings for the amount refused were taken after 1 hour. The percent of the fodder taken was determined using Eq 1:

$$Fodder\ consumed = \frac{(Fodder\ offered - Fodder\ refused)}{Fodder\ offered}\ x\ 100 \qquad (1)$$

The relative palatability ranks were then assigned to each species based on the percentage of fodder consumed by the heifers.

## Farmers' preference

A survey was conducted in the nearby three villages to learn about farmers' preferences for the fodder species. Ten farmers were chosen from each village and asked to rate the fodder species chosen for the current study, as well as their preferred time of fodder harvest.

## Statistical analysis

Seasonal variability analysis was performed for the nutritive analysis of selected traits i.e., DM, CP, EE, total ash, NFE, OM, total carbohydrate, P, K, Ca, Cu, Fe, Mn, Zn, CF, ADF, NDF, total phenol, tannin, nitrate, HCN and saponin contents of different MPTs. The data for nutritive parameters and palatability were statistically analysed using the analysis of variance (ANOVA) of a factorial randomized block design ignoring the missing value, as described by Gomez and Gomez [48]. SAS data analysis package 9.2 was used to test the mean of the treatments for significance at the 5% level of significance, and graphs were created using JMP 15.1 and R software v. 4.0.5. In addition to it, a multivariate analysis was carried out to assess the variations in the nutritional and anti-nutritional contents as a function of the 15 MPTs. For this analysis, the palatability, proximate composition, mineral contents, cell wall constituents and anti-nutritional contents for the respective MPTs were analyzed through a principal component analysis (PCA) based on the Pearson correlation coefficient index using the XLSTAT 202.5.1 software.

## Results

### Proximate composition

The proximate composition of the selected 15 MPTs revealed that there were significant variations among MPTs and during different seasons (Tables 2 and 3). The maximum DM content was recorded in *A. catechu* (62.50%) followed by *Q. leucotrichophora* (59.69%), and the lowest (34.22%) in *F. roxburghii*, which was found to be statistically equivalent to *L. leucocephala* (35.02%). The CP content ranged from the 8.49% (*A. catechu*) to 19.40% (*A. chinensis*) with an average of 13.30%. The ether content averaged 3.87%, varying from 2.32 (*M. composita*) to 6.55% (*A. catechu*). The highest mean ash content was recorded in *C. australis* (19.59%), while lowest was in *A. chinensis* (4.93%), the average being 10.40%. The maximum and minimum NFE was observed in *P. floribundum* (60.43%) and *A. chinensis* (33.49%), respectively. *A. chinensis* had the highest OM content (95.07%), whereas the minimum was found in *C. australis* (80.41%), which also recorded the minimum total carbohydrate content (61.45%), while the maximum was detected in *Q. leucotrichophora* (81.90%).

Similar to species effect, seasons also had a significant impact on the nutritional contents of the leaves, with the exception of NFE content (Tables 2 and 3). In the autumn season, the maximum dry matter content (53.07%) was detected, followed by the winter season, while the minimum was in the spring season (41.68%), however it remained statistically similar to summer season. The highest CP content was recorded in the spring season leaves (15.35%) and the lowest in winter (10.75%). While, CF (20.58–28.94%) and carbohydrate contents (69.77–75.78%) had an opposite trend. The maximum EE content (3.93%) was recorded in the winter, while the minimum was in summer season (3.38%) leaves. The ash content increased from spring (9.19%) to autumn (11.31%) and decreased during winter season (lowest value, 8.75%). The OM content was maximum and minimum during winter (91.25%) and autumn (88.82%), respectively.

**Table 2. Proximate analysis of MPT leaves in relation to season of harvesting (n = 36).**

| Multipurpose Tree Species (MPTs) | Dry matter content (%) | | | | | Crude protein content (%) | | | | | Carbohydrate content (%) | | | | | Ether extract (%) | | | | |
|---|---|---|---|---|---|---|---|---|---|---|---|---|---|---|---|---|---|---|---|---|
| | $S_1$ | $S_2$ | $S_3$ | $S_4$ | Mean | $S_1$ | $S_2$ | $S_3$ | $S_4$ | Mean | $S_1$ | $S_2$ | $S_3$ | $S_4$ | Mean | $S_1$ | $S_2$ | $S_3$ | $S_4$ | Mean |
| A. catechu | 61.44 | - | 63.32 | 62.72 | 62.50[h] | 6.50 | - | 11.08 | 7.87 | 8.49[a] | 74.66 | - | 76.58 | 74.97 | 75.40[efg] | 7.53 | - | 5.15 | 6.96 | 6.55[g] |
| A. chinensis | 27.90 | 38.56 | 48.22 | 58.73 | 43.35[c] | 28.00 | 26.25 | 13.42 | 9.92 | 19.40[f] | 63.11 | 66.95 | 79.06 | 84.26 | 73.34[cde] | 4.02 | 2.37 | 2.25 | 1.95 | 2.65[ab] |
| B. variegata | - | 63.23 | 52.72 | 34.73 | 50.23[e] | - | 11.08 | 9.04 | 6.12 | 8.75[a] | - | 84.80 | 78.99 | 77.23 | 80.34[gh] | - | 3.64 | 5.13 | 5.34 | 4.70[ef] |
| C. australis | 33.55 | 52.26 | 58.33 | - | 48.05[de] | 27.42 | 9.33 | 9.92 | - | 15.56[de] | 52.97 | 71.21 | 60.17 | - | 61.45[a] | 3.51 | 3.02 | 3.66 | - | 3.40[bc] |
| F. roxburghii | 35.65 | 29.07 | 35.44 | 36.73 | 34.22[a] | 12.83 | 11.50 | 11.96 | 10.50 | 11.70[abc] | 65.09 | 73.44 | 64.87 | 70.00 | 68.35[bc] | 2.39 | 3.06 | 6.26 | 3.23 | 3.74[cd] |
| G. optiva | 52.87 | - | 57.62 | 55.63 | 55.37[f] | 14.0 | - | 22.5 | 21.0 | 19.16[ef] | 73.11 | - | 65.23 | 65.27 | 67.97[f] | 2.83 | - | 1.77 | 3.59 | 2.73[ab] |
| L. leucocephala | 32.77 | 29.74 | 42.92 | 34.64 | 35.02[a] | 21.00 | 22.75 | 17.33 | 16.33 | 19.35[ef] | 65.09 | 64.63 | 68.01 | 69.54 | 66.82[b] | 3.58 | 3.81 | 5.96 | 3.99 | 4.34[de] |
| M. composita | - | 36.23 | 48.13 | - | 42.18[c] | - | 18.08 | 14.00 | - | 16.04[def] | - | 69.75 | 83.29 | - | 69.01[bcd] | - | 1.93 | 2.71 | - | 2.32[a] |
| M. serrata | 42.58 | 30.42 | 44.97 | - | 39.32[b] | 15.17 | 17.50 | 12.25 | - | 14.97[cd] | 59.79 | 65.13 | 71.13 | - | 65.35[ab] | 4.08 | 4.34 | 4.56 | - | 4.33[de] |
| O. glandulifera | 52.76 | 44.85 | 59.59 | 47.17 | 51.09[e] | 14.58 | 9.92 | 9.32 | 8.17 | 10.50[ab] | 73.53 | 81.10 | 78.66 | 80.62 | 78.48[efgh] | 3.18 | 2.49 | 2.27 | 3.56 | 2.87[abc] |
| O. ojeinensis | - | 38.50 | 51.64 | 48.80 | 46.31[d] | - | 14.00 | 13.42 | 10.50 | 12.64[bcd] | - | 75.53 | 71.91 | 75.26 | 74.23[de] | - | 2.29 | 3.53 | 3.69 | 3.17[abc] |
| P. floribundum | 34.52 | 41.16 | 58.86 | 36.95 | 42.87[c] | 11.08 | 8.75 | 14.58 | 12.25 | 11.67[abc] | 70.69 | 70.90 | 76.41 | 78.08 | 74.02[def] | 4.78 | 7.19 | 3.47 | 3.90 | 4.84[ef] |
| Q. glauca | 51.10 | 52.58 | 56.74 | 59.18 | 54.90[f] | 7.58 | 14.00 | 9.92 | 8.75 | 10.06[ab] | 79.90 | 73.99 | 77.11 | 77.91 | 77.23[efgh] | 6.99 | 2.47 | 4.79 | 6.76 | 5.25[f] |
| Q. leucotrichophora | 58.47 | 51.04 | 63.41 | 65.85 | 59.69[g] | 8.17 | 7.00 | 12.25 | 10.50 | 9.48[ab] | 84.14 | 85.14 | 78.23 | 80.10 | 81.90[h] | 1.05 | 3.40 | 4.36 | 5.15 | 3.49[bcd] |
| S. tetrasperma | 16.52 | 46.78 | 54.17 | - | 39.16[b] | 14.00 | 11.08 | 9.92 | - | 11.67[ab] | 75.21 | 76.45 | 70.24 | - | 73.96[b] | 3.17 | 3.90 | 4.00 | - | 3.69[cd] |
| Mean | 41.68[a] | 42.65[a] | 53.07[c] | 49.19[b] | | 15.35[b] | 14.17[b] | 12.66[a] | 10.75[a] | | 69.77[a] | 73.77[bc] | 72.33[ab] | 75.78[c] | | 3.93[b] | 3.38[a] | 3.99[b] | 4.38[b] | |
| CD$_{0.05}$ | MPTs = 2.69; S = 1.39; | | | | | MPTs = 3.82; S = 1.97; | | | | | MPTs = 5.02; S = 2.59; | | | | | MPTs = 0.89; S = 0.46; | | | | |
| | MPTs × S = 1.78 | | | | | MPTs × S = 2.55 | | | | | MPTs × S = 3.35 | | | | | MPTs × S = 0.59 | | | | |

$S_1$- Spring (March-April); $S_2$- Summer (May-June); $S_3$- Autumn (September-November); $S_4$- Winter (December-February).
Means followed by different letters are significantly different (P<0.05)

**Table 3. Proximate analysis of MPT leaves in relation to season of harvesting (n = 36).**

| Multipurpose Tree Species (MPTs) | Total ash content (%) | | | | | Nitrogen free extract (%) | | | | | Organic matter content (%) | | | | |
|---|---|---|---|---|---|---|---|---|---|---|---|---|---|---|---|
| | $S_1$ | $S_2$ | $S_3$ | $S_4$ | Mean | $S_1$ | $S_2$ | $S_3$ | $S_4$ | Mean | $S_1$ | $S_2$ | $S_3$ | $S_4$ | Mean |
| A. catechu | 11.31 | - | 7.18 | 10.20 | 9.56[bcd] | 50.07 | - | 57.85 | 52.43 | 53.45[ef] | 88.69 | - | 92.82 | 89.80 | 90.44[def] |
| A. chinensis | 4.87 | 4.43 | 5.28 | 5.16 | 4.93[a] | 29.78 | 31.48 | 34.15 | 38.55 | 33.49[a] | 95.13 | 95.57 | 94.72 | 94.84 | 95.07[g] |
| B. variegata | - | 7.76 | 6.68 | 6.34 | 6.98[ab] | - | 59.39 | 49.13 | 45.89 | 51.47[cde] | - | 92.24 | 93.16 | 93.66 | 93.02[fg] |
| C. australis | 16.10 | 16.43 | 26.25 | - | 19.59[g] | 35.57 | 47.59 | 35.89 | - | 39.68[ab] | 83.90 | 83.57 | 73.75 | - | 80.41[a] |
| F. roxburghii | 19.69 | 12.00 | 16.91 | 16.27 | 16.22[f] | 50.51 | 57.54 | 44.38 | 48.21 | 50.16[cde] | 84.31 | 88.00 | 83.09 | 83.73 | 83.78[b] |
| G. optiva | 9.71 | - | 10.51 | 9.83 | 10.02[cde] | 54.86 | - | 43.65 | 47.80 | 48.77[cde] | 89.92 | - | 89.49 | 90.17 | 89.86[cd] |
| L. leucocephala | 10.33 | 8.81 | 8.52 | 10.14 | 9.45[bc] | 53.94 | 53.41 | 54.02 | 51.09 | 53.11[def] | 89.67 | 91.19 | 91.48 | 89.96 | 90.55[def] |
| M. composita | - | 10.23 | 14.81 | - | 12.52[e] | - | 57.44 | 53.72 | - | 55.58[ef] | - | 89.77 | 84.99 | - | 87.38[c] |
| M. serrata | 20.96 | 13.03 | 12.06 | - | 15.35[f] | 47.57 | 52.95 | 54.80 | - | 51.77[cde] | 79.04 | 86.97 | 87.94 | - | 84.65[b] |
| O. glandulifera | 8.71 | 6.49 | 9.75 | 7.65 | 8.15[bc] | 60.48 | 62.04 | 57.34 | 57.98 | 59.46[ef] | 91.29 | 93.51 | 90.25 | 92.35 | 91.85[def] |
| O. ojeinensis | - | 8.18 | 11.14 | 10.54 | 9.95[bcd] | - | 39.92 | 44.76 | 48.81 | 44.50[bcd] | - | 91.82 | 88.86 | 89.46 | 90.05[def] |
| P. floribundum | 9.95 | 9.66 | 9.03 | 9.27 | 9.48[bcd] | 56.95 | 54.27 | 64.93 | 65.56 | 60.43[ef] | 90.05 | 90.34 | 90.97 | 90.73 | 90.52[def] |
| Q. glauca | 5.52 | 9.54 | 8.18 | 6.59 | 7.46[b] | 36.32 | 34.06 | 37.46 | 36.63 | 36.12[a] | 94.48 | 90.46 | 91.82 | 93.42 | 92.55[efg] |
| Q. leucotrichophora | 6.64 | 4.39 | 7.36 | 4.25 | 5.66[a] | 55.60 | 54.84 | 42.84 | 22.25 | 43.88[bc] | 93.36 | 95.54 | 94.84 | 95.75 | 94.87[g] |
| S. tetrasperma | 7.62 | 8.57 | 15.84 | - | 10.68[de] | 58.71 | 58.36 | 47.99 | - | 55.02[ef] | 92.38 | 91.43 | 84.16 | - | 89.32[c] |
| Mean | 10.95[b] | 9.19[a] | 11.31[b] | 8.75[a] | | 49.19 | 51.02 | 48.19 | 46.83 | | 89.02[a] | 90.80[b] | 88.82[a] | 91.25[b] | |
| CD$_{0.05}$ | MPTs = 2.51; S = 1.30; | | | | | MPTs = 7.45; S = NS; | | | | | MPTs = 2.54; S = 1.31; | | | | |
| | MPTs × S = 1.68 | | | | | MPTs × S = 4.96 | | | | | MPTs × S = 1.69 | | | | |

$S_1$- Spring (March-April); $S_2$- Summer (May-June); $S_3$- Autumn (September-November); $S_4$- Winter (December-February).
Means followed by different letters are significantly different (P<0.05)

## Mineral composition

The mineral composition showed a significant (P<0.05) variation amongst different MPTs (Tables 4 and 5). The maximum P content was recorded in *B. variegata* and *M. serrata* leaves, each displaying identical values (0.25%), whereas the minimum P content was found in *A. catechu* (0.03%) leaves. Similarly, K and Ca contents also showed wide variation from 0.98% (*S. tetrasperma*) to 2.18% (*M. serrata*) and 10.53% (*C. australis*) and 1.66% (*A. chinensis*), respectively. The Cu and Fe contents showed less but significant variations ranging from 16.91 ppm (*M. serrata*) to 22.94 ppm (*A. catechu*) and 504.38 ppm (*O. glandulifera*) to 701.27 ppm (*A. chinensis*) (Table 5), respectively. *Q. glauca* had the highest Mn content (264.99 ppm), while *O. oojeinensis* (33.32 ppm) had the lowest. The minimum Zn content was detected in *A. catechu* (4.29 ppm), while the maximum was in *S. tetrasperma* (56.29 ppm) and thus displayed a huge variation amongst MPTs.

Seasonal variations also had a significant (P<0.05) effect on the mineral composition of the studied parameters, excepting K (Tables 4 and 5). The maximum P (0.20%), Ca (4.90%) and Zn (31.69 ppm) content was recorded in autumn season, and the minimum P (0.09%) and Zn (13.90 ppm) in winter season leaves. The highest Cu (22.89 ppm) and Fe (699.13 ppm) contents were recorded during winter season, while the lowest Cu (16.54 ppm) and Fe (653.84 ppm) contents were observed in summer and autumn, respectively. The K content did not change significantly with the seasons. However, the Mn content was the highest in spring (89.96 ppm) and the lowest in summer (60.51 ppm).

## Cell wall composition and anti-nutritional contents

There were significant differences (P<0.05) among MPTs for the cell wall constituents, i.e., ADF, NDF and CF contents (Table 6). Q. glauca had the highest ADF (43.79%) and NDF (61.13%), while *L. leucocephala* had the lowest ADF (12.18%) and NDF (21.51%). Similarly, *Q. glauca* had the maximum CF content (41.11%), while *M. composita* had the minimum (13.43%). Likewise, the anti-nutritional contents also varied significantly (P<0.05) among the different MPTs (Tables 6 and 7). The total tannin content varied from 0.57% (*M. serrata*) to 6.09% (*Q. glauca*). The HCN contents ranged from 0.0–0.08 mg 100 g$^{-1}$ DM under different MPTs. *M. composita* and *C. australis* recorded the maximum (15.85 ppm) and minimum (0.84 ppm) nitrate content, respectively. The saponin contents ranged from 5.40% (*G. optiva*) to 27.16% (*P. floribundum*), whereas phenol contents varied from 1.50% (*M. serrata*) to 15.36% (*L. leucocephala*).

The maximum ADF (29.76%), NDF (45.77%), and saponin contents (19.34%) were observed in winter season leaves, while the minimum ADF content was detected during summer season. Similarly, the minimum phenol contents were recorded in summer (5.39%). The highest tannin (4.60%) and HCN contents (0.03 mg/100 gm) were detected in spring, while the lowest tannin and HCN contents were recorded in summer (1.94%) and in autumn (0.00 mg 100 g$^{-1}$), respectively. The nitrate contents in MPTs' leaves showed an increasing trend from spring (5.70 ppm) to autumn (7.36 ppm) and then declined significantly in winter season (5.34 ppm). In *L. leucocephala*, the mimosine content followed an irregular pattern, with the highest levels (1.22%) recorded in the oldest leaves (spring season) and the lowest levels (0.80%) in the youngest leaves (winter season) (Fig 1).

## Relative Nutritional Value Index (RNVI) and relative palatability

The seasonal nutritional analysis revealed that *M. serrata* was the most nutritious MPT in the spring and summer seasons (Fig 2), while *A. chinensis* was the least nutritious in the spring and *O. glandulifera* in the summer season (S3–S6 Tables). Similarly, *G. optiva* was the most nutritious MPT and *Q. glauca* was the least nutritious in both the autumn and winter seasons.

**Table 4. Mineral composition of MPT leaves in relation to season of harvesting (n = 36).**

| Multipurpose Tree Species (MPTs) | Phosphorus content (%) | | | | | Potassium content (%) | | | | | Calcium content (%) | | | | | Copper content (ppm) | | | | |
|---|---|---|---|---|---|---|---|---|---|---|---|---|---|---|---|---|---|---|---|---|
| | S₁ | S₂ | S₃ | S₄ | Mean | S₁ | S₂ | S₃ | S₄ | Mean | S₁ | S₂ | S₃ | S₄ | Mean | S₁ | S₂ | S₃ | S₄ | Mean |
| A. catechu | 0.01 | - | 0.05 | 0.02 | 0.03[a] | 1.52 | - | 0.93 | 1.23 | 1.22[abc] | 5.05 | - | 3.66 | 4.70 | 4.47[de] | 25.90 | - | 18.40 | 24.53 | 22.94[h] |
| A. chinensis | 0.12 | 0.25 | 0.12 | 0.05 | 0.13[bcd] | 1.25 | 1.30 | 1.60 | 1.03 | 1.29[a] | 0.76 | 1.25 | 2.08 | 2.55 | 1.66[a] | 18.40 | 16.13 | 20.23 | 26.23 | 20.25[ef] |
| B. variegata | - | 0.21 | 0.41 | 0.12 | 0.25[f] | - | 1.90 | 1.20 | 0.00 | 1.03[ab] | - | 2.75 | 1.95 | 2.55 | 2.42[ab] | - | 18.40 | 26.23 | 22.70 | 22.44[gh] |
| C. australis | 0.18 | 0.18 | 0.29 | - | 0.22[ef] | 1.60 | 1.10 | 1.18 | - | 1.29[abc] | 9.28 | 11.93 | 10.38 | - | 10.53[i] | 14.00 | 17.50 | 21.00 | - | 17.50[ab] |
| F. roxburghii | 0.15 | 0.08 | 0.17 | 0.05 | 0.11[bc] | 1.10 | 1.65 | 1.60 | 1.80 | 1.54[abc] | 7.78 | 4.20 | 4.78 | 8.08 | 6.21[g] | 11.43 | 20.23 | 21.90 | 22.70 | 19.07[cde] |
| G. optiva | 0.23 | - | 0.21 | 0.17 | 0.20[def] | 1.30 | - | 1.83 | 2.23 | 1.79[abc] | 6.05 | - | 4.15 | 2.63 | 4.28[de] | 15.63 | - | 21.83 | 21.03 | 19.50[de] |
| L. leucocephala | 0.15 | 0.06 | 0.30 | 0.07 | 0.15[cde] | 1.78 | 1.80 | 1.97 | 2.25 | 1.95[abc] | 3.20 | 3.75 | 3.73 | 5.68 | 4.09[de] | 16.60 | 17.60 | 22.83 | 21.80 | 19.71[e] |
| M. composita | - | 0.18 | 0.26 | - | 0.22[ef] | - | 1.85 | 0.68 | - | 1.26[abc] | - | 5.75 | 9.75 | - | 7.75[h] | - | 16.63 | 18.40 | - | 17.52[ab] |
| M. serrata | 0.50 | 0.14 | 0.11 | - | 0.25[f] | 2.33 | 1.60 | 2.62 | - | 2.18[c] | 3.25 | 3.43 | 5.35 | - | 4.01[de] | 14.93 | 16.60 | 19.20 | - | 16.91[ab] |
| O. glandulifera | 0.06 | 0.15 | 0.16 | 0.17 | 0.14[cd] | 1.60 | 1.20 | 1.83 | 1.80 | 1.61[abc] | 3.08 | 3.15 | 4.68 | 3.71 | 3.65[cd] | 15.67 | 14.90 | 23.63 | 26.20 | 20.10[ef] |
| O. ojeinensis | - | 0.14 | 0.28 | 0.11 | 0.18[cdef] | - | 1.70 | 1.68 | 1.05 | 1.48[abc] | - | 3.48 | 5.48 | 5.32 | 4.76[e] | - | 15.70 | 17.60 | 21.00 | 18.10[abc] |
| P. floribundum | 0.10 | 0.14 | 0.24 | 0.07 | 0.14[bcd] | 2.05 | 1.93 | 2.55 | 1.93 | 2.11[bc] | 3.78 | 2.98 | 3.10 | 4.38 | 3.56[cd] | 14.83 | 15.73 | 21.07 | 21.80 | 18.36[bcd] |
| Q. glauca | 0.05 | 0.12 | 0.05 | 0.03 | 0.06[ab] | 1.31 | 1.45 | 0.65 | 1.40 | 1.20[abc] | 2.95 | 3.33 | 3.21 | 2.33 | 2.95[bc] | 13.10 | 16.60 | 19.23 | 22.70 | 17.91[abc] |
| Q. leucotrichophora | 0.02 | 0.07 | 0.23 | 0.12 | 0.11[b] | 1.13 | 1.50 | 1.33 | 1.23 | 1.30[abc] | 2.73 | 2.00 | 2.88 | 3.78 | 2.85[bc] | 13.07 | 15.70 | 21.00 | 21.07 | 17.71[ab] |
| S. tetrasperma | 0.10 | 0.17 | 0.24 | - | 0.17[cdef] | 0.90 | 1.25 | 0.78 | - | 0.98[a] | 2.43 | 4.35 | 8.30 | - | 5.03[f] | 31.43 | 13.23 | 19.20 | - | 21.29[fg] |
| Mean | 0.14[b] | 0.15[b] | 0.20[c] | 0.09[a] | | 1.49 | 1.56 | 1.49 | 1.45 | | 4.19[a] | 4.03[a] | 4.90[b] | 4.15[a] | | 17.08[a] | 16.54[a] | 20.78[b] | 22.89[c] | |
| CD₀.₀₅ | MPTs = 0.08; S = 0.04; | | | | | MPTs = 1.12; S = NS; | | | | | MPTs = 1.01; S = 0.52; | | | | | MPTs = 1.22; S = 0.63; | | | | |
| | MPTs × S = 0.05 | | | | | MPTs × S = 0.75 | | | | | MPTs × S = 0.67 | | | | | MPTs × S = 0.81 | | | | |

S₁- Spring (March-April); S₂- Summer (May-June); S₃- Autumn (September-November); S₄- Winter (December-February).
Means followed by different letters are significantly different (P<0.05)

**Table 5. Mineral composition of MPT leaves in relation to season of harvesting (n = 36).**

| Multipurpose Tree Species (MPTs) | Iron content (ppm) | | | | | Manganese content (ppm) | | | | | Zinc content (ppm) | | | | |
|---|---|---|---|---|---|---|---|---|---|---|---|---|---|---|---|
| | $S_1$ | $S_2$ | $S_3$ | $S_4$ | Mean | $S_1$ | $S_2$ | $S_3$ | $S_4$ | Mean | $S_1$ | $S_2$ | $S_3$ | $S_4$ | Mean |
| A. catechu | 709.27 | - | 683.60 | 708.60 | 700.49[i] | 114.60 | - | 55.20 | 104.00 | 91.27[i] | 1.18 | - | 9.10 | 2.60 | 4.29[a] |
| A. chinensis | 691.90 | 671.07 | 737.70 | 704.40 | 701.27[i] | 31.40 | 27.20 | 38.67 | 37.67 | 33.73[a] | 16.40 | 5.10 | 8.80 | 5.60 | 8.98[b] |
| B. variegata | - | 687.80 | 696.10 | 687.87 | 690.59[g] | - | 38.60 | 24.10 | 63.60 | 42.10[bc] | - | 64.60 | 24.50 | 12.30 | 33.80[g] |
| C. australis | 625.27 | 679.40 | 691.93 | - | 665.53[c] | 38.60 | 46.90 | 30.30 | - | 38.60[b] | 26.67 | 18.50 | 12.60 | - | 19.26[d] |
| F. roxburghii | 646.10 | 687.83 | 691.50 | 700.20 | 681.41[e] | 91.60 | 38.60 | 73.63 | 77.60 | 70.36[g] | 4.33 | 16.67 | 5.30 | 1.17 | 6.87[ab] |
| G. optiva | 658.87 | - | 687.80 | 696.10 | 680.92[e] | 77.00 | - | 36.60 | 64.60 | 59.40[e] | 61.40 | - | 65.70 | 37.70 | 54.93[h] |
| L. leucocephala | 654.53 | 675.30 | 712.73 | 691.90 | 683.62[f] | 33.47 | 41.70 | 44.90 | 37.70 | 39.44[bc] | 16.97 | 9.33 | 50.30 | 2.90 | 19.88[de] |
| M. composita | - | 683.67 | 675.47 | - | 679.57[d] | - | 42.80 | 25.10 | - | 33.95[a] | - | 16.90 | 39.50 | - | 28.20[f] |
| M. serrata | 687.80 | 675.30 | 729.40 | - | 697.50[h] | 64.60 | 41.77 | 41.37 | - | 49.24[d] | 12.63 | 16.67 | 22.00 | - | 17.10[cd] |
| O. glandulifera | 658.60 | 650.30 | 0.00 | 708.60 | 504.38[a] | 53.20 | 25.10 | 38.27 | 41.77 | 39.58[bc] | 9.80 | 1.60 | 9.60 | 0.00 | 5.25[a] |
| O. ojeinensis | - | 671.07 | 683.60 | 691.90 | 682.19[e] | - | 27.20 | 38.27 | 34.50 | 33.32[a] | - | 8.30 | 12.30 | 7.20 | 9.27[b] |
| P. floribundum | 650.37 | 679.43 | 691.90 | 696.20 | 679.48[d] | 41.70 | 39.70 | 47.00 | 44.97 | 43.34[c] | 18.80 | 27.20 | 26.90 | 54.40 | 31.83[g] |
| Q. glauca | 658.57 | 671.10 | 704.47 | 704.40 | 684.63[f] | 206.80 | 157.00 | 578.60 | 117.57 | 264.99[k] | 5.10 | 63.80 | 10.50 | 12.10 | 22.88[e] |
| Q. leucotrichophora | 654.50 | 650.30 | 708.60 | 700.27 | 678.42[d] | 258.80 | 188.10 | 105.10 | 109.20 | 165.30[j] | 13.10 | 9.10 | 23.10 | 16.90 | 15.55[c] |
| S. tetrasperma | 600.40 | 658.67 | 712.77 | - | 657.28[b] | 67.70 | 71.90 | 56.30 | - | 65.30[f] | 7.80 | 5.97 | 155.10 | - | 56.29[h] |
| Mean | 658.01[b] | 672.40[c] | 653.84[a] | 699.13[d] | | 89.96[d] | 60.51[a] | 82.23[c] | 66.65[b] | | 16.18[b] | 20.29[c] | 31.69[d] | 13.90[a] | |
| $CD_{0.05}$ | MPTs = 1.33; S = 0.69; | | | | | MPTs = 4.63; S = 2.39; | | | | | MPTs = 3.57; S = 1.84; | | | | |
| | MPTs × S = 0.89 | | | | | MPTs × S = 3.29 | | | | | MPTs × S = 2.38 | | | | |

$S_1$- Spring (March-April); $S_2$- Summer (May-June); $S_3$- Autumn (September-November); $S_4$- Winter (December-February).

Means followed by different letters are significantly different (P<0.05)

Table 6. Cell-wall composition and total tannin content of MPT leaves in relation to season of harvesting (n = 36).

| Multipurpose Tree Species (MPTs) | Acid detergent fiber content (%) | | | | | Neutral detergent fiber (%) | | | | | Crude fiber content (%) | | | | | Total tannin content (%) | | | | |
|---|---|---|---|---|---|---|---|---|---|---|---|---|---|---|---|---|---|---|---|---|
| | S$_1$ | S$_2$ | S$_3$ | S$_4$ | Mean | S$_1$ | S$_2$ | S$_3$ | S$_4$ | Mean | S$_1$ | S$_2$ | S$_3$ | S$_4$ | Mean | S$_1$ | S$_2$ | S$_3$ | S$_4$ | Mean |
| A. catechu | 23.75 | - | 34.70 | 28.17 | 28.87[e] | 51.00 | - | 39.95 | 44.11 | 45.02[f] | 24.59 | - | 18.73 | 22.54 | 21.95[b] | 12.07 | - | 9.95 | 11.73 | 11.25[g] |
| A. chinensis | 21.20 | 22.46 | 23.52 | 25.94 | 23.28[cd] | 32.30 | 34.37 | 39.88 | 48.81 | 38.84[d] | 33.33 | 35.48 | 44.91 | 45.71 | 39.86[d] | 5.91 | 2.27 | 4.30 | 0.59 | 3.27[d] |
| B. variegata | - | 24.63 | 28.37 | 33.07 | 28.69[e] | - | 36.75 | 41.37 | 46.50 | 41.54[e] | - | 25.41 | 29.86 | 31.35 | 28.87[c] | - | 2.25 | 3.47 | 4.23 | 3.32[d] |
| C. australis | 21.72 | 22.22 | 27.87 | - | 23.94[cd] | 36.63 | 39.14 | 41.25 | - | 39.01[d] | 17.40 | 23.62 | 24.29 | - | 21.77[b] | 3.25 | 0.25 | 1.66 | - | 1.80[bc] |
| F. roxburghii | 31.73 | 32.87 | 35.16 | 36.23 | 34.00[f] | 41.25 | 42.92 | 50.88 | 53.63 | 47.17[g] | 14.58 | 15.90 | 20.49 | 21.79 | 18.19[ab] | 6.78 | 0.35 | 4.13 | 5.86 | 4.28[e] |
| G. optiva | 25.50 | - | 16.20 | 23.23 | 21.64[c] | 43.26 | - | 25.79 | 36.12 | 35.06[c] | 18.23 | - | 21.57 | 17.77 | 19.19[ab] | 4.50 | - | 0.14 | 2.61 | 2.42[c] |
| L. leucocephala | 14.29 | 10.30 | 11.67 | 12.46 | 12.18[a] | 23.47 | 18.59 | 21.86 | 22.10 | 21.51[a] | 11.15 | 11.22 | 14.16 | 18.45 | 13.75[a] | 1.46 | 2.55 | 3.16 | 0.56 | 1.93[bc] |
| M. composita | - | 11.11 | 14.68 | - | 12.89[a] | - | 20.00 | 23.72 | - | 21.86[a] | - | 12.31 | 14.55 | - | 13.43[a] | - | 2.40 | 4.96 | - | 3.68[d] |
| M. serrata | 14.47 | 17.59 | 21.27 | - | 17.77[b] | 24.50 | 27.93 | 30.35 | - | 27.59[b] | 12.22 | 12.19 | 16.32 | - | 13.58[a] | 1.11 | 0.49 | 0.10 | - | 0.57[a] |
| O. glandulifera | 21.22 | 26.73 | 33.27 | 35.93 | 29.29[e] | 56.01 | 58.56 | 60.55 | 67.23 | 60.59[i] | 13.05 | 19.06 | 21.32 | 22.65 | 19.02[ab] | 2.58 | 4.04 | 5.31 | 6.78 | 4.68[e] |
| O. ojeinensis | - | 22.64 | 27.00 | 25.79 | 25.15[d] | - | 33.13 | 37.09 | 44.22 | 38.15[d] | - | 35.61 | 27.15 | 26.45 | 29.74[c] | - | 1.38 | 2.58 | 0.78 | 1.58[b] |
| P. floribundum | 13.23 | 23.30 | 17.37 | 13.41 | 16.83[b] | 29.13 | 34.92 | 23.00 | 24.00 | 27.76[b] | 13.74 | 16.63 | 11.49 | 12.52 | 13.59[a] | 0.96 | 0.22 | 2.93 | 2.35 | 1.62[bc] |
| Q. glauca | 50.33 | 29.99 | 46.37 | 48.47 | 43.79[g] | 65.08 | 55.91 | 61.16 | 62.35 | 61.13[i] | 43.59 | 39.94 | 39.65 | 41.28 | 41.11[d] | 7.06 | 3.44 | 9.53 | 4.33 | 6.09[f] |
| Q. leucotrichophora | 38.89 | 42.43 | 43.49 | 44.67 | 42.37[g] | 50.24 | 51.66 | 52.16 | 54.45 | 52.13[h] | 28.54 | 30.30 | 35.39 | 57.85 | 38.02[d] | 6.83 | 5.12 | 7.29 | 4.60 | 5.96[f] |
| S. tetrasperma | 21.60 | 32.94 | 35.77 | - | 30.10[e] | 36.08 | 47.28 | 51.77 | - | 45.04[f] | 16.49 | 18.09 | 22.25 | - | 18.94[ab] | 2.42 | 0.48 | 1.70 | - | 1.53[b] |
| Mean | 25.17[a] | 24.55[a] | 27.50[b] | 29.76[c] | | 40.75[b] | 38.55[a] | 40.05[b] | 45.77[c] | | 20.58[a] | 22.75[ab] | 24.14[b] | 28.94[c] | | 4.60[c] | 1.94[a] | 4.08[b] | 4.04[b] | |
| CD$_{0.05}$ | MPTs = 2.95; S = 1.52; | | | | | MPTs = 1.85; S = 0.96; | | | | | MPTs = 5.81; S = 3.00; | | | | | MPTs = 0.84; S = 0.43; | | | | |
| | MPTs × S = 1.96 | | | | | MPTs × S = 1.24 | | | | | MPTs × S = 3.87 | | | | | MPTs × S = 0.56 | | | | |

S$_1$- Spring (March-April); S$_2$- Summer (May-June); S$_3$- Autumn (September-November); S$_4$- Winter (December-February).

Means followed by different letters are significantly different (P<0.05)

**Table 7. Anti-nutritional analysis of MPT leaves in relation to season of harvesting (n = 36).**

| Multipurpose Tree Species (MPTs) | Phenol content (%) | | | | | Nitrate content (ppm) | | | | | Saponin content (%) | | | | | Hydrocyanic acid (mg/100g DM) | | | | |
|---|---|---|---|---|---|---|---|---|---|---|---|---|---|---|---|---|---|---|---|---|
| | $S_1$ | $S_2$ | $S_3$ | $S_4$ | Mean | $S_1$ | $S_2$ | $S_3$ | $S_4$ | Mean | $S_1$ | $S_2$ | $S_3$ | $S_4$ | Mean | $S_1$ | $S_2$ | $S_3$ | $S_4$ | Mean |
| A. catechu | 15.33 | - | 13.08 | 12.58 | 13.67[h] | 0.00 | - | 15.27 | 0.00 | 5.09[c] | 27.68 | - | 22.04 | 28.06 | 25.93[i] | 0.00 | - | 0.06 | 0.00 | 0.02[ab] |
| A. chinensis | 10.13 | 3.57 | 5.97 | 8.97 | 7.16[de] | 2.70 | 0.87 | 6.30 | 0.00 | 2.47[ab] | 11.72 | 15.84 | 21.38 | 12.64 | 15.39[efg] | 0.10 | 0.00 | 0.00 | 0.00 | 0.03[b] |
| B. variegata | - | 7.60 | 11.38 | 15.18 | 11.39[g] | - | 6.08 | 0.00 | 12.50 | 6.19[c] | - | 15.15 | 12.42 | 14.41 | 13.99[cdef] | - | 0.00 | 0.00 | 0.00 | 0.00[a] |
| C. australis | 5.73 | 1.43 | 2.75 | - | 3.30[ab] | 1.33 | 1.20 | 0.00 | - | 0.84[a] | 9.52 | 13.55 | 12.21 | - | 11.76[cde] | 0.10 | 0.15 | 0.00 | - | 0.08[d] |
| F. roxburghii | 11.21 | 3.49 | 6.18 | 6.84 | 6.93[de] | 0.00 | 6.20 | 9.43 | 6.13 | 5.44[cd] | 20.93 | 9.39 | 12.41 | 7.70 | 12.61[def] | 0.01 | 0.00 | 0.00 | 0.11 | 0.03[b] |
| G. optiva | 4.80 | - | 2.90 | 4.40 | 4.03[bc] | 14.53 | - | 3.13 | 0.13 | 5.93[c] | 2.25 | - | 8.73 | 5.21 | 5.40[a] | 0.00 | - | 0.00 | 0.00 | 0.00[a] |
| L. leucocephala | 10.10 | 21.23 | 22.20 | 7.92 | 15.36[h] | 3.23 | 4.28 | 0.00 | 0.00 | 1.88[ab] | 18.10 | 13.52 | 15.17 | 20.71 | 16.87[fgh] | 0.02 | 0.00 | 0.00 | 0.00 | 0.01[a] |
| M. composita | - | 3.41 | 8.49 | - | 5.95[cd] | - | 14.20 | 17.50 | - | 15.85[g] | - | 5.43 | 10.88 | - | 8.16[abc] | - | 0.12 | 0.00 | - | 0.06[a] |
| M. serrata | 2.71 | 1.42 | 0.37 | - | 1.50[a] | 0.00 | 7.63 | 7.43 | - | 5.02[c] | 16.24 | 2.64 | 2.17 | - | 7.02[ab] | 0.02 | 0.00 | 0.00 | - | 0.01[a] |
| O. glandulifera | 3.52 | 5.29 | 6.98 | 14.31 | 7.53[de] | 0.00 | 24.50 | 2.73 | 0.00 | 6.81[de] | 23.36 | 8.55 | 17.90 | 25.00 | 18.70[gh] | 0.00 | 0.00 | 0.00 | 0.00 | 0.00[a] |
| O. ojeinensis | - | 2.70 | 4.64 | 3.62 | 3.65[abc] | - | 5.37 | 0.00 | 0.00 | 1.79[ab] | - | 11.17 | 24.45 | 26.41 | 20.68[h] | - | 0.00 | 0.00 | 0.00 | 0.00[a] |
| P. floribundum | 3.54 | 1.32 | 7.21 | 5.86 | 4.40[bc] | 4.99 | 0.00 | 0.73 | 21.43 | 6.79[de] | 23.22 | 30.12 | 22.90 | 32.40 | 27.16[i] | 0.00 | 0.00 | 0.00 | 0.00 | 0.00[a] |
| Q. glauca | 9.42 | 4.19 | 12.68 | 7.26 | 8.39[ef] | 0.00 | 19.33 | 12.07 | 0.00 | 7.85[e] | 4.79 | 10.10 | 19.30 | 24.14 | 14.58[cdefg] | 0.11 | 0.00 | 0.00 | 0.13 | 0.06[c] |
| Q. leucotrichophora | 9.37 | 6.07 | 10.26 | 6.23 | 7.98[de] | 14.43 | 1.10 | 28.03 | 18.53 | 15.53[g] | 4.69 | 6.58 | 14.12 | 16.10 | 10.37[bcd] | 0.00 | 0.00 | 0.00 | 0.00 | 0.00[a] |
| S. tetrasperma | 12.76 | 8.29 | 10.69 | - | 10.58[fg] | 27.23 | 3.50 | 7.70 | - | 12.81[f] | 17.07 | 30.48 | 13.23 | - | 20.26[h] | 0.00 | 0.00 | 0.00 | - | 0.00[a] |
| Mean | 8.22[a] | 5.39[b] | 8.39[a] | 8.47[a] | | 5.70[a] | 7.25[b] | 7.36[b] | 5.34[a] | | 14.96[a] | 13.27[a] | 15.29[a] | 19.34[b] | | 0.03[c] | 0.02[b] | 0.00[a] | 0.02[b] | |
| CD$_{0.05}$ | MPTs = 2.31; S = 1.19; | | | | | MPTs = 1.42; S = 0.74; | | | | | MPTs = 4.43; S = 2.29; | | | | | MPTs = 0.02; S = 0.01; | | | | |
| | MPTs × S = 1.54 | | | | | MPTs × S = 0.95 | | | | | MPTs × S = 2.95 | | | | | MPTs × S = 0.02 | | | | |

$S_1$- Spring (March-April); $S_2$- Summer (May-June); $S_3$- Autumn (September-November); $S_4$- Winter (December-February).

Means followed by different letters are significantly different (P<0.05)

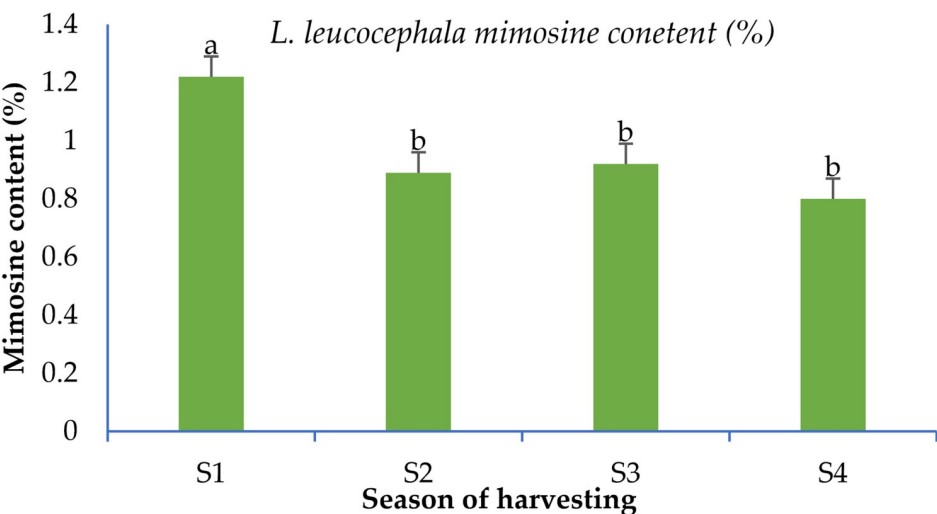

**Fig 1. Mimosine content (%) in *Leucaena leucocephala* during different seasons.** S1- Spring (March-April); S2- Summer (May-June); S3- Autumn (September-November); S4- Winter (December-February). Means followed by different letters are significantly different (P<0.05). The error bar signifies standard error of mean.

Simultaneously, the palatability of MPTs' leaves varied with the fodder tree species (Table 8). *L. leucocephala* was the most palatable one (97.86%) and followed by the other species in the order: *B variegata > G. optiva > M. serrata > A. catechu > P. floribundum > O. oojeinensis > C. australis > O. ferruginea > S. tetrasperma > F. roxburghii > A. chinensis > Q. glauca > Q. leucotrichophora > M. composita* (38.47%).

## Principal component analysis (PCA)

The spatial representation of the different nutritive and anti-nutritive values through PCA demonstrated the distinctness of different MPTs (Fig 3). The principal axes 1 and 2 obtained

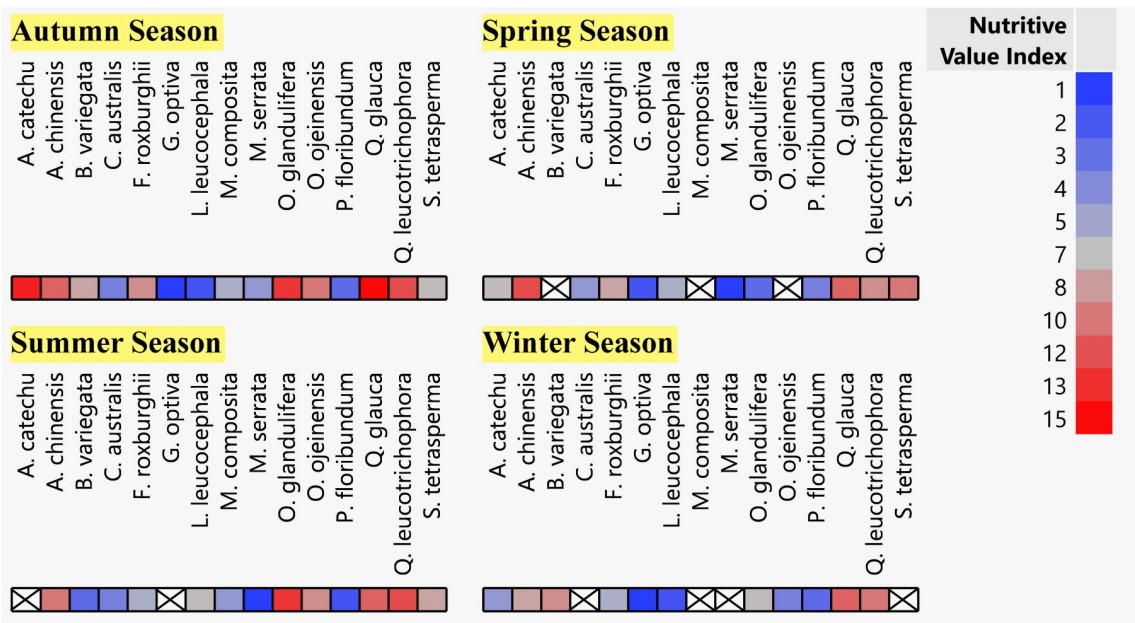

**Fig 2. Ranking of 15 selected fodder tree species during different season for relative nutritional value index.**

**Table 8. Relative palatability and ranking of different fodder tree species.**

| Species | Days of Feeding (D) | | | | | | | Rank |
|---|---|---|---|---|---|---|---|---|
| | $D_1$ | $D_2$ | $D_3$ | $D_4$ | $D_5$ | $D_6$ | Mean | |
| *A. catechu* | 94.67 | 95.17 | 94.67 | 95.67 | 94.67 | 95.33 | 95.03[e] | 5 |
| *A. chinensis* | 85.00 | 80.83 | 78.33 | 86.67 | 81.67 | 85.00 | 82.92[cd] | 12 |
| *B. variegata* | 94.17 | 98.33 | 98.33 | 98.33 | 96.67 | 95.83 | 96.94[e] | 2 |
| *C. australis* | 91.67 | 90.00 | 90.00 | 90.00 | 90.00 | 91.67 | 90.56[de] | 8 |
| *F. roxburghii* | 80.83 | 88.33 | 79.17 | 86.67 | 78.33 | 88.33 | 83.61[cd] | 11 |
| *G. optiva* | 96.67 | 97.50 | 96.67 | 95.00 | 95.00 | 93.33 | 95.69[e] | 3 |
| *L. leucocephala* | 97.50 | 98.00 | 97.50 | 98.50 | 97.67 | 98.00 | 97.86[e] | 1 |
| *M. composita* | 45.83 | 54.17 | 22.50 | 41.67 | 29.17 | 37.50 | 38.47[a] | 15 |
| *M. serrata* | 98.33 | 95.00 | 95.83 | 94.17 | 95.00 | 93.33 | 95.28[e] | 4 |
| *O. glandulifera* | 91.83 | 88.33 | 78.33 | 85.00 | 83.33 | 78.33 | 84.19[cd] | 10 |
| *O. ojeinensis* | 91.67 | 91.67 | 93.33 | 93.33 | 94.17 | 92.50 | 92.78[e] | 7 |
| *P. floribundum* | 95.00 | 94.00 | 93.33 | 91.67 | 92.50 | 93.67 | 93.36[e] | 6 |
| *Q. glauca* | 86.67 | 51.67 | 86.67 | 93.33 | 72.50 | 77.50 | 78.06[bc] | 13 |
| *Q. leucotrichophora* | 63.33 | 65.00 | 61.67 | 79.67 | 80.00 | 77.50 | 71.19[b] | 14 |
| *S. tetrasperma* | 75.00 | 80.00 | 95.00 | 93.33 | 76.67 | 86.67 | 84.44[cd] | 9 |
| **Mean** | 85.88 | 84.53 | 84.09 | 88.20 | 83.82 | 85.63 | | |
| **CD$_{0.05}$** | S = 7.57; D = non-significant; S×D = non-significant | | | | | | | |

Means followed by different letters are significantly different (P<0.05)

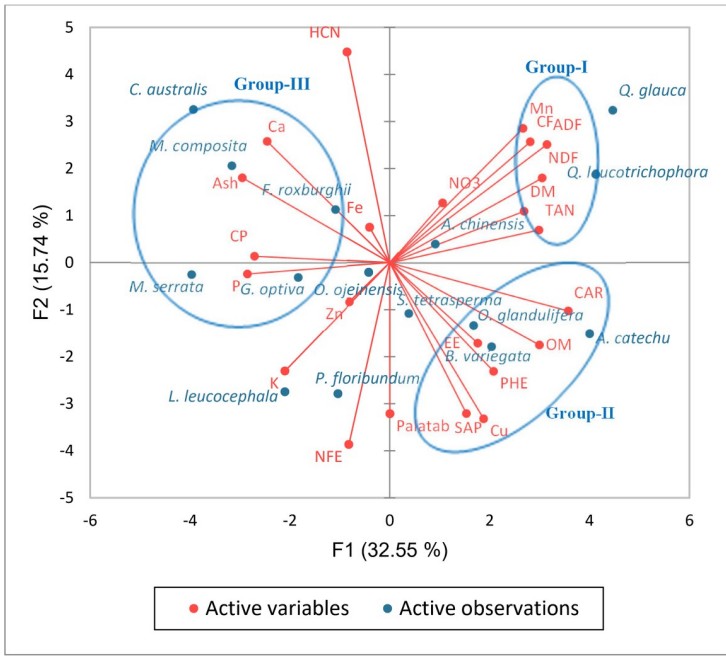

**Fig 3. Principle component analysis showing the palatability, proximate composition, mineral contents, cell wall constituents and anti-nutritional contents in 15 MPTs.** DM—dry matter (%); CP–crude protein (%); EE–ether extract; CF—crude fiber (%); Ash–total ash (%); ADF—acid detergent fiber (%); NDF—neutral detergent fiber (%); NFE—nitrogen free extract (%); OM—organic matter (%); CAR–carbohydrate (%); P = phosphorus (%); K = potassium (%); Ca—Calcium (%); Cu—copper (ppm); Fe—iron (ppm); Mn—manganese (ppm); Zn—zinc (ppm); PHE—phenol (%); TAN—total tannin (%); HCN—Hydrocyanic acid (mg 100g$^{-1}$); NO3—Nitrate (ppm); SAP—saponin content (ppm); Palatab–palatability.

in the analysis accounted for the 32.55 and 15.74% of total variation, respectively (cumulative value = 48.29%). The PCA classified the various nutritive and anti-nutritive values into three major groups, i.e., group-I Mn, CF, ADF, NDF, DM, Tannin, group-II Carbohydrate, OM, EE, PHE, Cu, SAP; and group-III Ca, Ash, CP and P while, the remaining parameters and palatability were quite distant. *Q. glauca* and *Q. leucotricophora* showed the highest contents of the group-I nutritive parameters, and low K and Zn contents. *L. leucocephala* and *P. floribundum* shared the common characteristics and a high K and NFE contents. *A. catechu*, *B. variagta* and *O. glandulifera* were rich in the group–II nutritive parameters but low in group-III nutritive parameters, while the opposite was detected for *M. serrata*, *C. australis* and *F. roxburghii*, which were rich in group-III nutritive parameters.

## Farmers' preference

Farmer preference rating of fifteen MPTs of mid-hill ecosystem of north-western Himalayas is presented in Table 9. Amongst different MPTs farmer preferred *G. optiva* the most, followed by *B. variegata* > *M. serrata* > *P. floribundum* > *O. oojeinensis* > *C. australis* > *L. leucocephala* > *Q. glauca* > *Q. leucotrichophora* > *A. catechu* > *A. chinensis* > *F.roxburghii* > *S. tetrasperma* > *O. glandulifera* and *M. composita*. Farmers preferred to harvest the leaves of *A. catechu*, *B. variegata*, *F. roxburghii*, *G. optiva*, *O. oojeinensis* and *P. floribundum* as fodder during the winter season, while *A. chinensis*, *C. australis*, *Q. glauca* and *Q. leucotrichophora* are fed to animal during the summer season. *M. serrata* and *S. tetrasperma* are harvested and offered to livestock during the spring season, whereas *L. leucocephala* could be harvested and fed to animals at any time of year. However, most of the farmers did not feed *M. composita* and *O. glandulifera* to their livestock regularly, so optimum harvesting time could not be ascertained.

## Discussion

### Proximate and mineral composition

The present research demonstrated that the nutritional values and palatability of the fifteen different MPTs investigated varied significantly among themselves and on a seasonal basis. The values obtained for most of the proximate composition parameters evaluated were

**Table 9. Preferred ranking and harvesting time for the fodder species by farmers for animal feeding.**

| Species | Ranking | Harvesting time |
| --- | --- | --- |
| *A. catechu* | 10 | November- February |
| *A. chinensis* | 11 | April-May |
| *B. variegata* | 2 | November–December |
| *C. australis* | 6 | April–May |
| *F. roxburghii* | 12 | September or November-December |
| *G. optiva* | 1 | November- February |
| *L. leucocephala* | 7 | Year around when available |
| *M. composita* | 15 | - |
| *M. serrata* | 3 | March-November |
| *O. glandulifera* | 14 | December-February |
| *O. ojeinensis* | 5 | May-December |
| *P. floribundum* | 4 | November- February |
| *Q. glauca* | 8 | April-May |
| *Q. leucotrichophora* | 9 | April-May |
| *S. tetrasperma* | 13 | May-June |

consistent with those obtained in prior studies [16, 49–53]. However, there are still discrepancies between the present and previous literature values, owing to the intrinsic genotype [18], edapho-climatic factors [22], management regimes [23], and other factors. Similarly, seasonal changes influence the composition of forage nutrients, which has an effect on the feed intake, digestibility, and energy released by farm animals after consumption [54].

Generally, lower temperature in the winter season has a detrimental effect on the growth of plants. Moreover, the scarce rainfall and other climatic conditions tend to affect the photosynthetic process, resulting in lower forage yield and proximate and mineral composition changes [55]. In addition, in the present investigation, it has been observed that the leaf phenology also played a major role. Specifically, except for the CF, the majority of the proximate and mineral parameter content of the MPTs increased as leaves matured [12], which is consistent with the current investigation. The DM was the highest during the autumn season (53.07%), which is in accordance with Gonzalez-Garcia and Archimede [56]. The CP contents of MPTs foliage was comparable to previous studies [57–59] and declined as the season proceeded from summer to winter, i.e., from younger to mature leaves. This may be attributed to the dilution effect, which happens when nutrients (particularly N) are redistributed to other plant parts at the end of the growth cycle [24]. Globally, many leguminous tree species are used as cattle feed, mostly because of their higher protein content throughout the year [60, 61]. However, in the present study, two leguminous tree species, *i.e.*, *A. catechu and B. variegata*, along with *Q. leucotrichophora*, possessed a CP content lower than 10%, whereas all other fodder tree species had a CP content greater than 10%, which is beneficial for rumen fermentation [62]. Therefore, despite belonging to the Fabaceae family, *A. catechu* and *B. variegata* reported a considerably low CP content, indicating that the proximate composition can largely depend on individual species rather than on family characteristics. For instance, previous studies in Algeria reported higher CP content [63], while similar values were found in Europe [64] compared to the present investigation. The EE content of the leaves was found to be considerably lower (2–6%) than the optimum requirement (S7 Table) and previous studies (up to 7.60%) [9]. Furthermore, contrarily to Shaheen et al. [65], in the present research, the ash content varied according to the species and corroborated well with earlier findings [9, 50, 66], but it was not influenced by seasonality. MPTs under investigation contained an average OM concentration around 90%, similarly to previous findings [16, 67]. The carbohydrate concentrations ranged from 61–82 percent and increased with leaves, which in divergence with the observations of Singh & Todaria [24], which can be owed to variance in NFE and CF contents.

In the present study, the mineral compositions varied significantly with the variation in the species and season. These differences in mineral composition among the tree species can be owed to differences in agro-climatic zones, maturity level, genetic makeup, season, soil fertility and harvest technique [50]. The P content (0.03–0.25%) in the present study was consistent with the level reported by Ganai et al. [9]; Singh et al. [50]; Gonzalez-Garcia et al. [56]. The highest P concentration occurred during the autumn season, which is comparable to the values reported earlier [5, 24, 25, 68]. However, Ca content in the present study was found to be higher than that reported in the literature (1.95–6.31% Ca) [50, 69–71] and increased as the growing season progressed [72] or at leaf maturity [73]. For this reason, the Ca: P ratios in the present study were considerably wide and varied from 1:16 to 1:149, compared to 2:1 indicating efficient utilization [74], thus highlighting P deficiency in most of the MPTs of the mid-hill Himalayan ecosystem.

Contrary to previous findings [25, 70, 75], the K contents in the present investigations did not vary significantly across seasons. The Cu contents in *A. catechu*, *A. chinensis*, *B. variegata*, *L. leucocephala*, *M. composita*, *M. serrata*, *O. glandulifera*, *O. oojeinensis*, *P. floribundum*, and *Q. glauca* increased with time, which might be related to copper immobility in plants [76]. The

decrease in Cu contents of *G. optiva*, *Q. leucotrichophora*, and *S. tetrasperma* with the maturity of leaves can be explained by the dilution effect caused by the rapid growth of their leaves [77]. Similar to Ca content, Fe content in present evaluation (500–700 ppm) was found to be consistent with earlier findings (520–801 ppm by Shinde and Sankhyan [78]; Rawat et al. [79]; Mahieu et al. [64]) or higher (133.05–467 ppm) [16, 64, 80], increased with maturity [72] and peaking in winter season [25, 75].

Mn contents in the MPTs in the present study was higher (33.73–264.99 ppm) than previously reported values by other workers (34.10–90.38 ppm [16, 66, 78]). The zinc contents of *A. catechu*, *A. chinensis*, *B. variegata*, *G. optiva*, *Q. glauca*, and *Q. leucotrichophora* was observed to decrease with leaves maturity. This may be because Zn can be mobilized from old reserves for photosynthesis, and the decrease in Zn concentrations in later stages may be due to the dilution effect caused by the rapid growth of leaves during this period [77]. Similarly, Yan et al. [81] found that Zn content was higher at the start of the growing season and decreased as the season progressed. Further, MPTs likes, *B. variegata*, *G. optiva*, *P. floribundum*, and *S. tetrasperma* were observed to possess sufficient zinc level to fulfil the dietary needs of the dairy cattle (S7 Table), while other species were zinc deficient.

## Cell wall composition and anti-nutritional contents

As the season progressed, ADF and NDF levels were found to increase, due to an increase in leaf cellulose, hemicellulose, and lignin contents [4, 19]. The ADF content of tree leaves in the present investigation (12–43%) is consistent with previous worker's values (22–49%) [29, 49, 50, 58], whereas the NDF was marginally higher (211–61%) than that stated by Khan et al. [57], Singh et al. [50], Muhammad et al. [82], and Mhaiskar [83] for different fodder species (24.26–41.77%). Similar to ADF and NDF contents, the CF content increased from young to mature leaves; since CF is composed of the ADF and NDF fractions. Also, Anele et al. [84] observed a general increase in leaf lignification in mature leaves, resulting in an increase in CF content.

Furthermore, chemical compounds, likes phenol, tannin, mimosine, HCN, nitrate and saponin contents were also assessed, as they are known to play an important role in animal health and productivity, either directly or through their metabolic products, or they can diminish nutrient intake, digestion, absorption, utilization, and produce ill consequences [85]. The values of the phenol content in the current study (1–13%) were consistent with the concentrations reported in *A. nilotica* (16.2%), *B. variegata* (4.8%), *O. oojeiuealis* (4.2%) and *L. leucocephala* (4.9%) [7], *Celtis africana* (1.4%) [86] and *Quercus* spp. (7–10%) [67]. Similarly, the tannin content was also comparable with Rana et al. [7] (2.1–14.6%), Adeduntan and Oyerinde [87] (0.3–053%), and Raju et al. [88]. Under the season effect, the phenol and total tannin contents increased with leaf maturity in *A. catechu*, *B. variegata*, *G. optiva*, *L. leucocephala*, *M. composita*, *O. glandulifera*, and *O. oojeinensis*, except in *A. chinensis*, *C. australis*, *F. roxburghii*, *M. serrata*, *P. floribundum*, and *Q. glauca*. This variation in species effect may be a result of physiological behavior and genetic makeup, leading to differential seasonal changes in their phenol and tannin contents [89]. The increase in phenol contents under different species during the winter may be a defensive mechanism against herbivorous insect attacks. Similarly, the increased tannin content in spring may be due to tannin condensation during the winter season, as this is an adaptive mechanism of frost resistant mesophyll cells designed to avoid injury during unfavorable temperate conditions [90] and to protect newly emerging leaves from herbivorous insect attacks.

The HCN concentrations measured in our fodder samples (up to 0.08 mg 100g$^{-1}$) are lower than those documented by various researchers (0.03–2.14 mg 100g$^{-1}$ [5, 87, 91]. Seasonal variation in the nitrate content of the MPTs leaves may be assigned to increased nitrate absorption

from the soil during the active growing season when temperatures were higher. However, the saponin contents (5–27%) were higher than those observed in *L. leucocephala* (5.8%) by Aye and Adegun [92]. Additionally, *L. leucocephala* leaves contains a non-protein amino acid called mimosine, which gets converted into dihydroxypyridone in rumen and can cause excessive salivation, hair loss, poor growth and swelling thyroid in livestock [16]. The mimosine content in the leaves of *L. leucocephala* was found to be vary from 0.80–1.22%, which is lower than previously recorded (0.8–2.9%; [93]). Therefore, in the livestock feed, the *L. leucocephala* leaves should be no more than 30 per cent of total feed on dry mater basis [16]. Further, the majority of the nutritional values in the MPTs of the mid-hills Himalayan fodder tree species were observed to be within the dairy cattle's optimum tolerable concentration range (S7 Table). However, Q. glauca's CF content, especially during the winter season, exceeded the maximum tolerable level for dairy cattle.

## Relative palatability and farmers' preference

The highest palatability (%) amongst the studied MPTs based on dry matter intake was observed with *L. leucocephala*, a leguminous tree, while the lowest was observed with *M. composita*, which is in consonance with the observations of Gunasekharan et al [94]. The better palatability of *L. leucocephala* can be attributed to the presence of secondary plant metabolites, such as beta-carotene and xanthophylls [95]. Further, *G. optiva* was observed to be the most nutritious MPT species in both autumn as well as in winter seasons and also the most favored species among farmers as well. *P. floribundum* received a higher farmer ranking and also have high palatability although this species has sporadic occurrence in the region. Farmers favored *Q. glauca* and *Q. leucotrichophora* over *A. chinensis* and *A. catechu*, believing that these trees provided animals with complete contentment. This will allow their use as supplements to low-quality fodder and straw-based diets in ruminants in order to improve animal health and milk productivity. Indeed, there is urgent need for establishing large scale plantations of highly nutritious and palatable species, like *G. optiva*, *L. leucocephala*, *B. variegata*, *M. serrata*, and *P. floribundum* on farmlands, common lands and wastelands.

## Conclusions

The present study concludes that there are significant variations among the fifteen different MPTs of the mid-hills north-western Himalayan ecosystem in the proximate and mineral compositions, cell wall constituents, anti-nutrient content, and palatability, which are also influenced by the seasonal effect. Except for EE, Ca, Cu, OM, and carbohydrate contents, the majority of the nutritive contents (CP, total ash, NFE) and mineral composition (P, K, Fe, Mn, Zn) decreased as leaves matured, while cell wall constituents (CF, ADF, NDF) and anti-nutritional content (total phenol, tannin, nitrate, HCN and saponin content) increased. Moreover, the majority of MPTs were found to be high in CP but low in EE. However, there were differences in terms of nutritive value, palatability, and farmer preference. For nutritive rich fodder for the livestock during the spring season, the preference should be accorded to leaves of *M. serrata*, *G. optiva*, *O. glandulifera*, *P floribundum* and *C. australis*, whereas in summer season *M. serrata*, *P floribundum*, *B. variegata* and *C. australi*s are the preferred ones. Similarly, in the autumn season the usage should be shifted toward *G. optiva*, *L. leucocephala*, *P floribundum*, *C australis* and *M. serrata*, while, in winter season *G. optiva*, *L. leucocephala*, *P floribundum*, *O. oojeinensis* and *A. catechu* could be the better choice. Strictly, due to the higher CF content, *Q. glauca* should be ignored for livestock feeding in the winter season. *L. leucocephala* is the most palatable, while *M. composita* was the most unappealing. Thus, MPTs forage harvested at the optimal stage of maturity has significant potential as a source of high-quality forage for

livestock, even during critical periods. The finding will help animal nutritionists, policymakers and ecologists to take appropriate measures for the year-round production of nutritive fodder, as well as the conservation and propagation of selected MPTs in sufficient quantity in a variety of agroforestry systems. The current study focused exclusively on the nutritive value of the prominent MPTs of the mid-hill Himalayan ecosystem, further investigation is also required to determine how different management practices, such harvesting intensity, could be optimized to produce quality fodder. Simultaneously, more emphasis should be given on nutritive analysis based on trees of different age groups as well as to in-vivo research trials able to identify suitable tree species for livestock production sustainability.

## Supporting information

**S1 File. Protocol for chemical analysis followed in current investigation.**
(DOCX)

**S1 Table. Month of palatability trial for fodder tree species.**
(DOCX)

**S2 Table. Scheme of establishment of palatability trial.**
(DOCX)

**S3 Table. Ranking of fodder tree species during spring season for nutritional value.**
(DOCX)

**S4 Table. Ranking of fodder tree species during summer season for nutritional value.**
(DOCX)

**S5 Table. Ranking of fodder tree species during autumn season for nutritional value.**
(DOCX)

**S6 Table. Ranking of fodder tree species during winter season for nutritional value.**
(DOCX)

**S7 Table. Nutrient and mineral requirement for dairy cattle (DM basis).**
(DOCX)

## Acknowledgments

The authors are grateful to the Head of the Department of Silviculture and Agroforestry, Y.S. Parmar, University of Horticulture and Forestry, Solan (HP), India, for providing the necessary facilities during the study.The authors also duly acknowledge the use of the facilities provided by AICRP on Agroforestry of YSPUH&F centre.

## Author Contributions

**Conceptualization:** Manasi Rajendra Navale, D. R. Bhardwaj, Rohit Bishist, C. L. Thakur.

**Formal analysis:** Manasi Rajendra Navale, D. R. Bhardwaj, Subhash Sharma, Prashant Sharma.

**Investigation:** Manasi Rajendra Navale, D. R. Bhardwaj, Rohit Bishist, C. L. Thakur, Subhash Sharma.

**Methodology:** Manasi Rajendra Navale, D. R. Bhardwaj, Rohit Bishist, C. L. Thakur.

**Visualization:** Prashant Sharma.

**Writing – original draft:** Prashant Sharma.

**Writing – review & editing:** Dhirender Kumar, Massimiliano Probo.

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
