## [Decision Letter · Decision Letter 0]

4 Aug 2022

PONE-D-22-16121Seasonal variations in the nutritive value of fifteen multipurpose fodder tree species: a case study of north-western Himalaya mid-hillsPLOS ONE

Dear Dr. Sharma,

Thank you for submitting your manuscript to PLOS ONE. After careful consideration, we feel that it has merit but does not fully meet PLOS ONE’s publication criteria as it currently stands. The reviewers has recommended publication and my own reading of your MS concurs with this view. However, the reviewers has suggested some further minor revisions to your MS. Therefore, we invite you to submit a revised version of the manuscript that addresses the points raised during the review process.

We look forward to receiving your revised manuscript.

Kind regards,

Sandeep Rawat, Ph.D.

Academic Editor

PLOS ONE

Journal Requirements:

Additional Editor Comments:

The MS on Seasonal variations in the nutritive value of fifteen multipurpose fodder tree species of Himalaya has scope of publication in this Journal, However, following points need special attention:

1. Introduction section is too long and can be reduced as also indicated by reviewer.

2. Much detailed methodology required for chemical investigation section.

3. Refine the conclusion section.

4. Justification of seasonal variation and classification of fodder species according to plant taxonomy will improve discussion.

5. Productivity and yield of fodder species in different seasons should be included.

Reviewers' comments:

**Comments to the Author**

1. Is the manuscript technically sound, and do the data support the conclusions?

Reviewer #1: Yes

Reviewer #2: Yes

2. Has the statistical analysis been performed appropriately and rigorously? 

Reviewer #1: Yes

Reviewer #2: Yes

3. Have the authors made all data underlying the findings in their manuscript fully available?

Reviewer #1: Yes

Reviewer #2: Yes

4. Is the manuscript presented in an intelligible fashion and written in standard English?

Reviewer #1: Yes

Reviewer #2: Yes

5. Review Comments to the Author

Reviewer #1: The manuscript is well written and well prepared.

However, authors have to advance discussion part to make findings important for international level and not only for african countries. The analytical values of feedstuffs are adequate to characterize before use.

Reviewer #2: Quality fodder to the livestock is the major problem in the country which requires immediate attention. The study by the authors assess and compare the different quality parameters and relative nutritive value of important fodder species of western Himalayan region. The present study is therefore fully justified and a good attempt in addressing the problem of fodder scarcity in Himalayan region. The manuscript s is well written and provides important results to be used by different stakeholders. The introduction is well-written, with clear justifications of background, research gaps, and rationale but need to be reduced. The methodology section has been explained in nice manner. Observations have been recorded in systematic manner using standard methods. Data has been properly analysed according to standard statistical procedure. Results and Discussion part needs some minor corrections as mentioned in track change mode. Table 1 need to be placed after L 126.

The manuscript can be submitted after incorporating the minor suggestions

6. PLOS authors have the option to publish the peer review history of their article (what does this mean?). If published, this will include your full peer review and any attached files.

Reviewer #1: No

Reviewer #2: **Yes**

[NOTE: Reviewer comments are submitted as an attachment file.]

---

## [Author Response · Author response to Decision Letter 0]

8 Sep 2022

First of all, we would like to thank the editor and reviewers for their constructive comments to improve the quality of the manuscript. The manuscript has been revised according to the suggestions received. Please find the revised manuscript as track mode change along with point-to-point response as follows: -

Editor

• Editor Comment: Introduction section is too long and can be reduced as also indicated by reviewer.

Authors Reply: The Introduction section has been reduced by deleting the following statements: 

However, the full potential of MPTs’ green fodder has not been completely exploited [9] to develop an alternative and healthy non-traditional feed for ruminants that would supply nutritious fodder throughout the year and increase livestock productivity, due to the paucity of adequate knowledge about their nutritive value and palatability.

Nevertheless, there is dearth of detailed information on the species used by farmers, their preference for exotic or indigenous MPTs, and the extent to which they are capable of supplementing and improving the nutritional value assessment in the laboratory. This approach will allow researchers to focus effectively and efficiently on farmers' objectives, while also providing them with results they can understand [28].

• Editor Comment: Much detailed methodology required for chemical investigation section.

Authors Reply: We have added a much more detailed description of the methodology used for chemical investigation in S1 File. Protocol for chemical analysis followed in current investigation.

• Editor Comment: Refine the conclusion section.

Authors Reply: Conclusion section has been revised thoroughly according to reviewers’ suggestions.

• Editor Comment: Justification of seasonal variation and classification of fodder species according to plant taxonomy will improve discussion.

Authors Reply: Compiled as per the comment and following statement added:

Generally, lower temperature in the winter season has a detrimental effect on the growth of plants. Moreover, the scarce rainfall and other climatic conditions tend to affect the photosynthetic process, resulting in lower forage yield and proximate and mineral composition changes [55]. In addition, in the present, investigation, it has been observed that the leaf phenology also played a major role.

Regarding the classification of the fodder species according to plant taxonomy, we did not find any significant link to highlight. For instance, A. catechu had the lowest crude protein, whereas A. chinensis the highest, but both species belong to the family Fabaceae. However, to better clarify, we have added / modified the following statement:

Globally, many leguminous tree species are used as cattle feed, mostly because of their higher protein content throughout the year [59]. However, in the present study, two leguminous tree species, i.e., A. catechu and B. variegata, along with Q. leucotrichophora, possessed a CP content lower than 10 %, whereas all other fodder tree species had a CP content greater than 10 %, which is beneficial for rumen fermentation [60]. Therefore, despite belonging to the Fabaceae family, A. catechu and B. variegata reported a considerably low CP content, indicating that the proximate composition can largely depend on individual species rather than on family characteristics.

• Editor Comment: Productivity and yield of fodder species in different seasons should be included.

Authors Reply: The average leaf dry biomass yield (kg DM tree-1yr-1) details have been added in the Table 1. 

 

Reviewer 1

• Reviewer Comment: However, authors have to advance discussion part to make findings important for international level and not only for african countries. The analytical values of feedstuffs are adequate to characterize before use.

Authors Reply: We try to include in the discussion comments and links to other international important papers important for Asian, European and American countries, even if there is still a limited number of studies on this topic, such as:

Mahieu S, Novak S, Barre P, Delagarde R, Niderkorn V, Gastal F, Emile JC. Diversity in the chemical composition and digestibility of leaves from fifty woody species in temperate areas. Agrofor Syst. 2021; 95: 1295-308. https://doi.org/10.1007/s10457-021-00662-2

Ammar H, López S, González JS, Ranilla MJ. Seasonal variations in the chemical composition and in vitro digestibility of some Spanish leguminous shrub species. Anim Feed Sci Technol. 2004; 115: 327-340. https://doi.org/10.1016/j.anifeedsci.2004.03.003

Ravetto Enri S, Probo M, Renna M, Caro E, Lussiana C, Battaglini LM, et al. Temporal variations in leaf traits, chemical composition and in vitro true digestibility of four temperate fodder tree species. Anim Prod Sci. 2020; 60: 643-658. https://doi.org/10.1071/AN18771

Kokten K, Kaplan M, Hatipoglu R, Saruhan V, Çinar S. Nutritive value of mediterranean shrubs. J Anim Plant Sci. 2012; 22: 188-194.

Tolera A, Khazaal K, Ørskov ER. Nutritive evaluation of some browse species. Anim Feed Sci Technol. 1997; 67: 181-195. https://doi.org/10.1016/S0377-8401(96)01119-4

Gonzalez-Garcia E, Caceres O, Archimede H. Nutritive value of edible forage from two Leucaena leucocephala cultivars with different growth habit and morphology. Agrofor Syst. 2009; 77: 131–141. https://doi.org/10.1007/s10457-008-9188-4

Fircks YO, Ericsson, T, Sennerby-Forsse L. Seasonal variation of macronutrients in leaves, stems and roots of Salix dasyclados Wimm. grown at two nutrient levels. Biomass Bioenergy 2001; 21: 321–334. https://doi.org/10.1016/S0961-9534(01)00045-9

Reviewer 2

• Reviewer Comment: Line 122 Rainy -R in lower case

Authors Reply: Corrected

• Reviewer Comment: Line 132 was not were

Authors Reply: Corrected

• Reviewer Comment: Line 152 Use uniform pattern for YSP

Authors Reply: Corrected

• Reviewer Comment: Line 271 L. leucocephala italics

Authors Reply: Corrected 

• Reviewer Comment: Line 300 Delete “The”

Authors Reply: Corrected 

• Reviewer Comment: Line 465-467, This paragraph is not concluded from the study and hence may be deleted

Authors Reply: According to your suggestion, the statement was removed from the manuscript.

• Reviewer Comment: Line 472-476 Conclusion is too long and hence some part may be taken to discussion section

Authors Reply: The above statement was removed from the conclusion section and added in the discussion part.

---

## [Decision Letter · Decision Letter 1]

12 Oct 2022

Seasonal variations in the nutritive value of fifteen multipurpose fodder tree species: a case study of north-western Himalaya mid-hills

PONE-D-22-16121R1

Dear Dr. Sharma,

We’re pleased to inform you that your manuscript has been judged scientifically suitable for publication and will be formally accepted for publication once it meets all outstanding technical requirements.

Kind regards,

Sandeep Rawat, Ph.D.

Academic Editor

PLOS ONE

Additional Editor Comments (optional):

Reviewers' comments:

Reviewer's Responses to Questions

**Comments to the Author**

1. If the authors have adequately addressed your comments raised in a previous round of review and you feel that this manuscript is now acceptable for publication, you may indicate that here to bypass the “Comments to the Author” section, enter your conflict of interest statement in the “Confidential to Editor” section, and submit your "Accept" recommendation.

Reviewer #2: All comments have been addressed

2. Is the manuscript technically sound, and do the data support the conclusions?

Reviewer #2: Yes

3. Has the statistical analysis been performed appropriately and rigorously? 

Reviewer #2: Yes

4. Have the authors made all data underlying the findings in their manuscript fully available?

Reviewer #2: Yes

5. Is the manuscript presented in an intelligible fashion and written in standard English?

Reviewer #2: Yes

6. Review Comments to the Author

Reviewer #2: (No Response)

7. PLOS authors have the option to publish the peer review history of their article (what does this mean?). If published, this will include your full peer review and any attached files.

Reviewer #2: **Yes: **Rajesh Kaushal

---

## [Editor Report · Acceptance letter]

17 Oct 2022

PONE-D-22-16121R1 

Seasonal variations in the nutritive value of fifteen multipurpose fodder tree species: a case study of north-western Himalayan mid-hills 

Dear Dr. Sharma:

I'm pleased to inform you that your manuscript has been deemed suitable for publication in PLOS ONE. Congratulations! Your manuscript is now with our production department. 

Kind regards, 

on behalf of

Dr. Sandeep Rawat 

Academic Editor

PLOS ONE